# Isolation and identification of a genotype F bovine enterovirus in western China

Kun Xu,[1,2,3] Xiaohan Wang,[1,2,3] Jie Yuan Guo,[1,2,3] Yanpei Ku,[1,2,3] Jiang Wang,[1,2,3] Beibei Chu,[1,2,3] Jiajia Pan,[1,2,3] Guoyu Yang[2,3,4]

**ABSTRACT** This study successfully isolated a novel bovine enterovirus strain from a bovine fecal sample, which was designated as Sichuan/SQ/20. The isolate showed typical enterovirus morphology under electron microscopy. Phylogenetic analysis showed that this strain exhibits the closest genetic relationship with the HeN-YR91 and JPN/TottoriU-31 strains, and all three belong to the BEV-F1 genosubtype. Subsequently, comprehensive investigations were conducted on the biological characteristics of this virus, both *in vivo* and *in vitro*. *In vitro* characterization revealed that viral replication commenced at 3 h post-infection (hpi) in Madin-Darby bovine kidney cells, reaching peak at 48 hpi with a virus titer of $1 \times 10^{8.73}$ $TCID_{50}$/0.1 mL. Cytopathic effects initially appeared at 12 hpi. A 12-minute treatment at 55°C was sufficient to completely inactivate the virus. *In vivo* analysis revealed that significant pathological changes were specifically observed in the spleen, with no lesions observed in other organs. Immunofluorescence assay detected specific fluorescent signals in the liver, spleen, and small intestine, which were consistent with the PCR results. These findings provide a scientific foundation for vaccine design and antiviral drug screening, as well as for the development of effective prevention and control strategies.

**IMPORTANCE** Bovine enterovirus (BEV) is an important pathogen causing calf diarrhea and has been detected in the feces of calves with diarrhea, although its pathogenicity remains unclear. This study systematically established an isolation and identification protocol for BEV, characterized its physicochemical properties, and further investigated the pathogenicity and tissue tropism of the isolated strain in mice. These findings establish crucial baseline data for future vaccine development and therapeutic intervention strategies.

**KEYWORDS** bovine enterovirus, virus isolate, biological characterization, pathogenicity

Bovine diarrhea syndrome remains a predominant cause of neonatal calf mortality and morbidity, with pathogenesis linked to triadic interactions of pathogens, environment, and host susceptibility. Although calf diarrhea has multi-factorial origins, viral etiologies are garnering escalating research attention due to their epidemic potential and evolutionary dynamics (1). Bovine enteric viruses constitute major diarrheal pathogens whose clinical significance is heightened through frequent coinfections that potentiate disease severity and complicate outbreak control (2, 3). Member viruses of this genus exhibit conserved biological properties and can cause respiratory and digestive tract diseases in animal hosts (4–8). Clinical manifestations include diarrhea, pyrexia, respiratory distress, and abortion in cattle, with severe cases progressing to fatal outcomes due to dehydrating enteritis and bronchopneumonia (4, 9, 10).

Bovine enterovirus (BEV) exhibits global distribution in cattle populations. BEV was detected at a rate of 14.5% in South American cattle herds surveyed between 2012

**Peer Reviewer** Eduardo Rodríguez-Román, Venezuelan Institute for Scientific Research, Caracas, Venezuela

Address correspondence to Jiajia Pan, first2015@163.com, or Guoyu Yang, haubiochem@163.com.

The authors declare no conflict of interest.

See the funding table on p. 12.

and 2016 (11). Epidemiological surveys conducted from 2021 to 2023 revealed a BEV prevalence range of 6.1%–10.7% in domestic cattle herds (3, 12). Serological surveillance of BEV across Turkish regions revealed a 64.8% antibody prevalence in cattle herds, indicating a high level of virus circulation (13). BEV was detected in 20.1% and 28% of fecal samples collected from cattle in Tunisia and Romania, respectively (14). BEV is primarily transmitted among cattle herds via the fecal-oral route. This transmission mode not only enhances viral dissemination efficiency but also complicates epidemic control measures, ultimately inflicting substantial economic losses on the cattle industry (5). The environmental persistence of viral particles in contaminated feed and water sources perpetuates transmission cycles, necessitating enhanced biosecurity measures and continuous monitoring.

BEV is a member of the enterovirus species in the Enterovirus genus within the Picornaviridae family. BEV is a small, non-enveloped, icosahedral particle containing a single-stranded, positive-sense viral RNA genome of approximately 7.5 kb in length (15). Currently, the Enterovirus genus consists of 15 species. Among these, the enteric viruses infecting cattle are Enterovirus E and Enterovirus F. EV-E has five subtypes (E1–E5), and EV-F has eight subtypes (F1–F8). The single-stranded RNA genomes of BEV contain a single open reading frame (ORF) that encodes a polyprotein comprising four structural proteins (VP1–VP4) and seven non-structural proteins (2A–2C and 3A–3D). VP1, along with VP2 and VP3, is exposed on the surface of the capsid. VP1 not only serves as the receptor-binding protein but also as the main neutralizing antigen. Therefore, the VP1 gene exhibits the highest sequence variability compared with other parts of the genome, which determines the classification of the enterovirus. All the non-structural proteins, including some of their precursor proteins, may be involved in the replication of the viral RNA.

The virus was first isolated by American scientist Moll in 1955 (16). In 2011, Yingli Li' group identified and isolated the bovine enteric virus in China, classifying the strain as enterovirus species F (17). Subsequently, BEV strains of different subtypes were isolated from various cell lines using multiple methods, offering diverse research approaches for the isolation and infection (5, 7, 18). In addition, BEV has been detected in sheep (19), camels (20), and other animals, with seropositivity confirming exposure and suggesting potential cross-species transmission. An increasing number of BEVs have been isolated, although most infections are subclinical and exhibit limited pathogenicity (21). Despite its characteristic high morbidity-low mortality profile, BEV demonstrates pathogenic potential through associations with respiratory, gastrointestinal, and occasional neurological manifestations (4, 22). Besides, as a member of the Picornaviridae family, BEV exhibits high mutation rates due to inherent characteristics of its genetic material and replication mechanisms. These frequent mutations potentially compromise vaccine efficacy, thereby posing a substantial threat to the cattle industry.

In this study, we collected 108 batches of cattle fecal samples from Sichuan, Qinghai, and Inner Mongolia from June to October 2024. Using conventional PCR identification and plaque purification methods, we isolated a bovine enteric virus F1 subtype strain, designated as Sichuan/SQ/20 (GenBank: PV290163.1). This study provides a comprehensive characterization of the Sichuan/SQ/20 strain. We investigated the growth kinetics, physicochemical properties, and genetic recombination patterns of this viral isolate. Concurrently, a mouse infection model was established to analyze BEV-induced pathological alterations in host organs and viral tissue distribution, thereby laying the groundwork for in-depth research on this pathogen. These findings establish crucial baseline data for future vaccine development and therapeutic intervention strategies.

## MATERIALS AND METHODS

### Sample collection and RT-PCR analysis

In this study, a total of 108 fecal and anal swab samples were collected from calves with diarrhea on large-scale cattle farms in three provinces, including Sichuan province (56), Qinghai province (21), and Inner Mongolia (31), from June to October 2024. All the samples were frozen in maintenance medium in virus-sampling tubes, transported to the laboratory within 2 days using cold chain transportation, and stored at −80°C.

A 20% fecal suspension was prepared and clarified in a sterile phosphate-buffered saline solution (PBS, pH = 7.2) by centrifugation of at $12,000 \times g$ for 20 min, followed by filtration through a 0.22 µm filter. Total RNA from fecal samples was extracted according to the manufacturer's instructions (the HiPure Viral RNA/DNA Midi Kit, Guangzhou Magen Biotechnology Co., Ltd.). The presence of the virus was confirmed by RT-PCR with the primers BEV-5′UTR-F and BEV-5′UTR-R (Table 1). The reaction system was performed using the Vazyme ClonExpress II One Step Cloning Kit (C112-01; Vazyme Biotech Co., Ltd.), following the manufacturer's instructions. The PCR was carried out under the following conditions: 50°C for 30 minutes, 95°C for 3 minutes, followed by 35 cycles of 95°C for 30 seconds, 55°C for 30 seconds, 72°C for 1 minute, and a final extension at 72°C for 10 minutes.

PCR products were subjected to electrophoresis with 1.0% agarose gel, visualized with a gel documentation system. The PCR product with 600 bp was confirmed by Sanger sequencing. The sequences were compared to existing database entries using the Basic Local Alignment Search Tool (BLAST).

### Cell culture

Madin-Darby bovine kidney cell (MDBK), Baby hamster kidney cell (BHK-21), and African green monkey kidney (Vero) cell were used to isolate the virus. These cells were cultured in Dulbecco's modified Eagle's medium (DMEM) supplemented with 10% fetal bovine serum (FBS) and 1% antibiotics at 37°C with 5% $CO_2$. The three types of cells were cultured in a six-well tissue culture plate, and virus isolation was carried out once the confluence reached 90%.

### Virus isolation

Five BEV-positive samples, each from a different region, were selected. Their supernatants were then repreared with antibiotics (100 U/mL penicillin and 100 µg/mL streptomycin), filtered through a 0.22 µm filter. Exactly 0.1 mL of the filtrate was diluted (1:5) and then inoculated with MDBK, BHK-21, and Vero cells, respectively. The monolayers of these cells, prepared in advance, were washed with PBS. The inoculum was discarded after incubation with the three cells for 2 h. The cells were washed with PBS before adding DMEM supplemented with 2% FBS. DMEM was added to the first well of the plate as a negative control. The culture was incubated at 37°C with 5% $CO_2$,

**TABLE 1** Target gene amplification PCR primer sequence in this study

| Fragment | Primer | Sequence (5′–3′) | Size (bp) | Application |
|---|---|---|---|---|
| 5′ | BEV-5′UTR_F | CCTTTGTACGCCTGTTTTCCC | 600 | Detection of BEV |
| | BEV-5′UTR_R | GAAACACGGAGTACCGAAAGTAGTC | | |
| 5′A | BEV-5′UTR_F | CCTTTGTACGCCTGTTTTCCC | 2,100 | Amplification of full genome sequence gene |
| | BEV-A_R | GTCATRAAAGTGCCDGTRAACATRCA | | |
| AE | BEV-C_F | TGYATGTTYACHGGCACTTTYATGAC | 2,200 | |
| | BEV-D_R | CAATGCGATTCTTGGTCTTGAACTGCA | | |
| E | BEV-E_F | TGCAGTTCAAGACCAAGAATCGCATTG | 1,900 | |
| | BEV-E_R | GGRAATGTRCARAGRGAYATGCC | | |
| F | BEV-F_F | ACTAGTGCAGGCTACCCCTATGT | 1,100 | |
| | BEV-F_R | CCAATTTGAATTATCCGGTCTAATCAGATTCTAATTGG | | |

and the cytopathic effect (CPE) on the cells was assessed daily. After five consecutive passages, viruses were harvested by three freeze-thaw cycles and were further confirmed by RT-PCR.

## Virus purification and identification

The isolated virus Sichuan/SQ/20 (PV290163) was purified by plaque purification. The MDBK cells were cultured in a six-well plate to form a monolayer with uniform distribution and a fusion rate of over 95%, washed with PBS, and then inoculated with the virus suspension that can cause cytopathic effects diluted to $10^{-3}$–$10^{-7}$ by DMEM without FBS. The plate was incubated at 37°C for 3 h. Afterward, the virus suspension was discarded. The cells were washed three times with PBS and recovered using a mixture of sterilized low-melting point agarose (4%) and 2× DMEM at a ratio of 1:1. Then, the cell culture plate was inverted and incubated at 37°C in a 5% $CO_2$. CPE were observed after 48 h and stained with neutral red. Single plaques were selected and placed in serum-free DMEM medium, then subjected to three freeze-thaw cycles. The mixture was then inoculated in the MDBK monolayer cultures in six-well tissue-culture plates. Finally, purified cultured virus samples were obtained after three rounds of plaque purification. To identify the purified strain, the cells were stained with 2 mL of a staining solution consisting of 0.5% crystal violet and 25% formaldehyde solution following the above procedure.

## Electron microscopy observation

MDBK cells were infected with the isolated Sichuan/SQ/20 strain and cultured. The virus supernatant was transferred into a dialysis bag (MD34), embedded in polyethylene glycol 8000, and concentrated at a ratio of 1:50. The concentrated and purified virus was obtained. The sample was examined by Transmission Electron Microscopy after being negatively stained with 2% phosphotungstic acid.

## Replication kinetics analysis of the isolates Sichuan/SQ/20

To determine the infectivity of the purified strain, titration of $TCID_{50}$ for Sichuan/SQ/20 isolates was performed using 96-well plates. The purified strain was serially diluted from $10^{-1}$ to $10^{-10}$ and used to infect the 96-well plates for each dilution at 37°C for 3 h after being washed with PBS. Forty-eight hours post-inoculation, record the cytopathic effects for each dilution and calculate the viral titer ($TCID_{50}$) using the Reed-Muench method. To detect virus growth kinetics, infected cells were harvested at 0, 3, 6, 9, 20, 24, 30, 48, and 60 h post-infection (hpi). The viral titer ($TCID_{50}$) was determined and calculated using the Reed-Muench method. To further examine temperature sensitivity, Sichuan/SQ/20 was heated at 4°C, 37°C, 42°C, 50°C, 54°C, 55°C, 56°C, and 60°C for 1 h. The viral titer ($TCID_{50}$) was determined and calculated using the Reed-Muench method.

To establish the minimum inactivation time at 55°C, the virus was diluted 1:10 in DMEM and distributed into five tubes (1 mL/tube). Following vortex mixing, tubes were heated at 55°C for durations ranging from 3 to 15 minutes. Viral titers ($TCID_{50}$) were then assessed for all samples.

## Immunofluorescence assay

To characterize the antigenic properties of the isolated strain, the reactivity between the Sichuan/SQ/20 strain and mouse positive serum was evaluated using an immunofluorescence assay (IFA). MDBK cell monolayers were infected with the Sichuan/SQ/20 strain at a multiplicity of infection of 0.1. At 24 hpi, the cells were washed three times by PBS, fixed with 4% paraformaldehyde for 30 min, and incubated with 0.1% Triton X-100 for 10 min. After three PBS washes, cells were blocked with 5% bovine serum albumin for 1 h, incubated with mouse positive serum for 1 h at room temperature, and uninfected cells served as the negative control. After three washes with PBS, the cells were probed with AF488-labeled goat anti-mouse IgG (H + L; 1:500; Beyotime, Cat# A0428) for 1 h at room temperature. After washing three times with PBS again, the cells were incubated with

4′,6-diamidino-2-phenylindole dihydrochloride (DAPI, Solarbio, Cat#C0065) for 10 min. Fluorescence images were tested by a microscope (Olympus, IX73, Tokyo, Japan).

## Virus genome amplification

Total RNA from the purified strain was extracted with the HiPure Viral RNA/DNA Midi Kit (Guangzhou Magen Biotechnology Co., Ltd.) following the manufacturer's instructions. The nearly complete genome sequence gene of the Sichuan/SQ/20 strain was obtained by RT-PCR with four pairs of primers (Table 1) using the HiScript II One Step RT-PCR Kit (Dye Plus) (Vazyme Biotech Co., Ltd.) following the manufacturer's instructions. The PCR products of the expected size, according to each set of primers, were purified using a Gel Extraction kit (TaKaRa, Dalian, China) after electrophoresis. The purified DNA was cloned into pMD18-T vector (TaKaRa, China), and the resulting plasmid was used to transform competent *E. coli* cells. Positive inserts were confirmed by PCR and further sequenced by Sangon Biotechnology Company (Shanghai, China). To prevent contamination, the preparation of the PCR mix and the addition of the template DNA were performed in separate rooms using dedicated pipets and filtered tips. Sequence assembly and manual editing were performed using the SeqMan program (DNASTAR, Madison, WI). The nucleotide sequence identities were calculated by the Megalign program available within the Lasergene software package (version 7.1, DNAstar).

## Phylogenetic analysis

The phylogenetic analysis of the Sichuan/SQ/20 strain was performed based on the nucleotide sequences of P1 and VP1 using the maximum likelihood (ML) method. ML trees were constructed using MEGA version 7.0, employing the general time-reversible (GTR) nucleotide substitution model with optimized parameters for the gamma (Γ)-distribution and the proportion of invariable sites (i.e., GTR + Γ + I). Bootstrap support values were calculated from 1,000 replicates performed in MEGA version 7.0 (23).

## Mice

All animal experiments were conducted in strict compliance with the Guide for the Care and Use of Laboratory Animals and the ethical guidelines of Henan Agricultural University. Six-week-old female BALB/c mice were purchased from Liaoning Changsheng Biotechnology Co., Ltd. The pathogenicity of the isolated Sichuan/SQ/20 strain was evaluated by inoculating BALB/c mice. The mice were randomly assigned to two groups (groups A and B; four mice per group) and housed in separate cage litter. Following a 3-day adaptation period, Group A mice ($n = 4$) were administered 0.5 mL of the viral isolate ($10^{7.66}$ TCID$_{50}$/0.1 mL) by intraperitoneal injection, and Group B mice ($n = 4$) served as negative controls and received an equivalent volume (0.2 mL) of sterile DMEM cell culture supernatant by intraperitoneal injection.

Clinical symptoms were monitored daily. During necropsy, spleen, liver, and small intestine tissues were collected and processed for dual purposes: one portion was snap-frozen at −80°C for viral RNA extraction and PCR detection, while the other portion was fixed in 4% paraformaldehyde for 24 h before undergoing paraffin embedding, sectioning, and staining for histological examination and IFA.

## RESULTS

### Detection and isolation of BEV in bovine stool samples

A total of 108 fecal and anal swab samples from three different provinces were detected by RT-PCR using the primer pair BEV-5′UTR_F and BEV-5′UTR_R (Table 1), which was designed based on the alignment results of the 5′UTR region of all BEVs available in the GenBank database. The results showed that BEV DNA was detected in six fecal samples. All BEV-specific RNA-positive fecal samples from three provinces were inoculated into MDBK, BHK-21, and Vero cells for BEV isolation. After five cycles of blind passage, a virus strain was isolated from positive fecal samples, which induced distinct CPE in MDBK cell

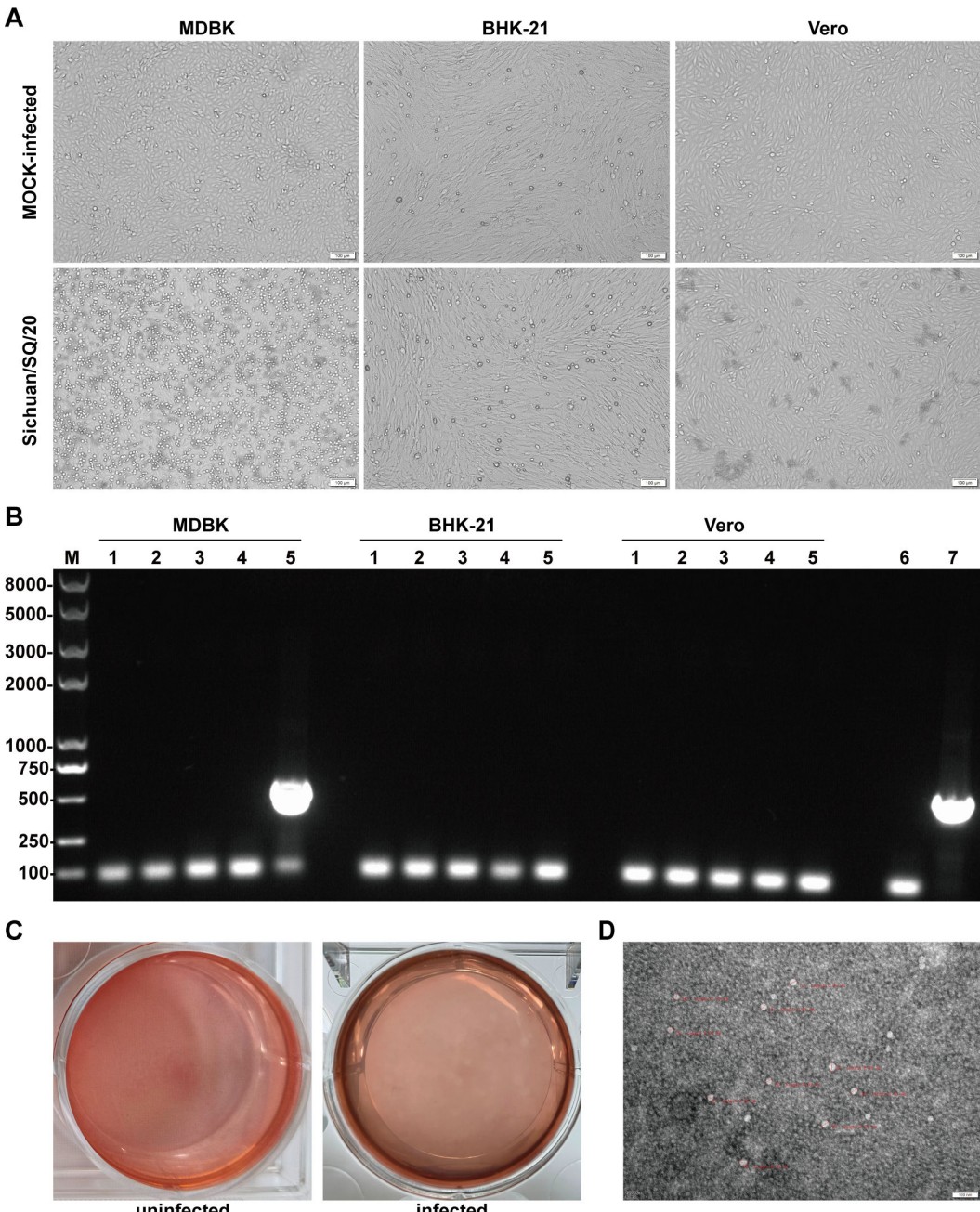

**FIG 1** Isolation and identification of BEV. (A) Morphology of MDBK, BHK-21, and Vero cells for BEV isolation. (B) PCR detection of samples after five consecutive passages. Lanes 1–5, five BEV-positive samples; lane 6, negative control; lane 7, positive control. (C) Plaque purification of the isolated virus. (D) Electron microscopic imaging of supernatants from Sichuan/SQ/20-inoculated MDBK cells using negative staining with phosphotungstic acid. Scale bar, 100 nm.

but not in BHK-21 and Vero cells (Fig. 1A). It was characterized by RT-PCR (Fig. 1B), and sequencing of the PCR products followed by online BLAST analysis revealed that the nucleotide sequence of the isolated strain was 100% identical to that of the published sequence (HeN-YR91), confirming that the isolated virus was BEV. Subsequently, MDBK cells were chosen for plaque purification, and clear, uniform plaques were obtained (Fig. 1C) after three rounds of purification. This purified strain was named "Sichuan/SQ/20 strain." The purified strain was observed using electron microscopy after negative staining, revealing that the virus particles were non-enveloped and spherical, with an average diameter of 30–40 nm (Fig. 1D), consistent with the size of picornaviruses.

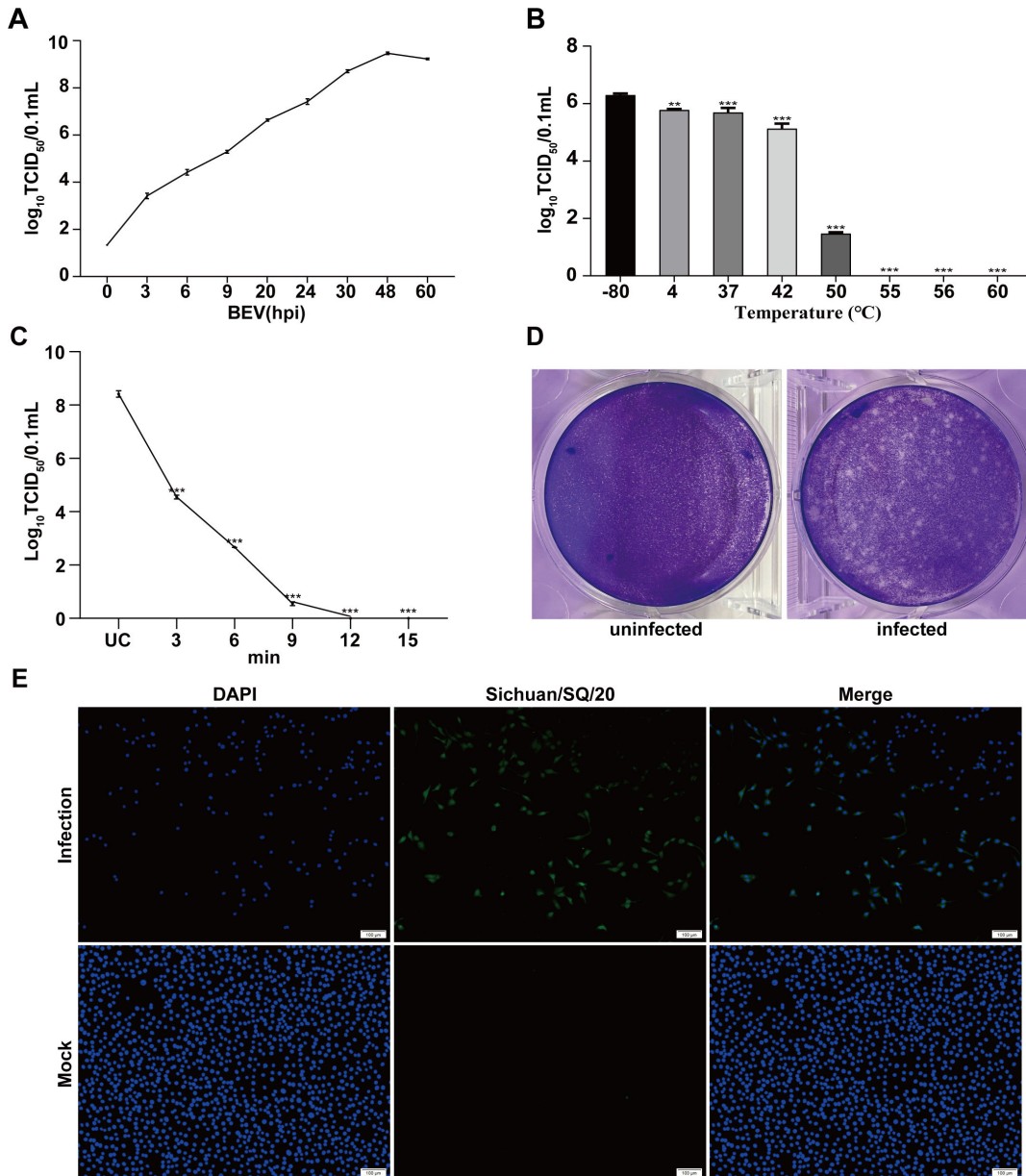

**FIG 2** Characterization of Sichuan/SQ/20 strain. (A) Growth kinetics of Sichuan/SQ/20 in MDBK (hpi). (B) Infectivity after thermal treatment at indicated temperatures (4°C–60°C). (C) Thermal inactivation kinetics at 55°C. (D) Representative plaque morphology under crystal violet staining. (E) MDBK cell post-BEV infection. Cell nuclei were visualized using 4′,6-diamidino-2-phenylindole under fluorescence microscopy. Scale bar = 100 µm. **$P < 0.01$, ***$P < 0.001$.

## Biological characteristics of the purified strain

The biological characteristics of the Sichuan/SQ/20 strain were further analyzed using MDBK cells. The titration of the purified strain was $10^{8.73} \times$ TCID$_{50}$/0.1 mL. Virus growth kinetics show that the virus titer presented a gradual upward tendency and peaked at 48 hpi (Fig. 2A). The Sichuan/SQ/20 strain exhibited thermal sensitivity with significantly reduced viability at 50°C (Fig. 2B). At temperatures exceeding 55°C, complete viral inactivation occurred within 12 minutes (Fig. 2C). Moreover, the purified virus exhibited well-defined plaque morphology (Fig. 2D). Specific immunofluorescence was detected in BEV-infected MDBK cells using hyperimmune serum from inactivated whole-virus vaccinated mice as primary antibody, while no signal was observed in uninfected control cells (Fig. 2E).

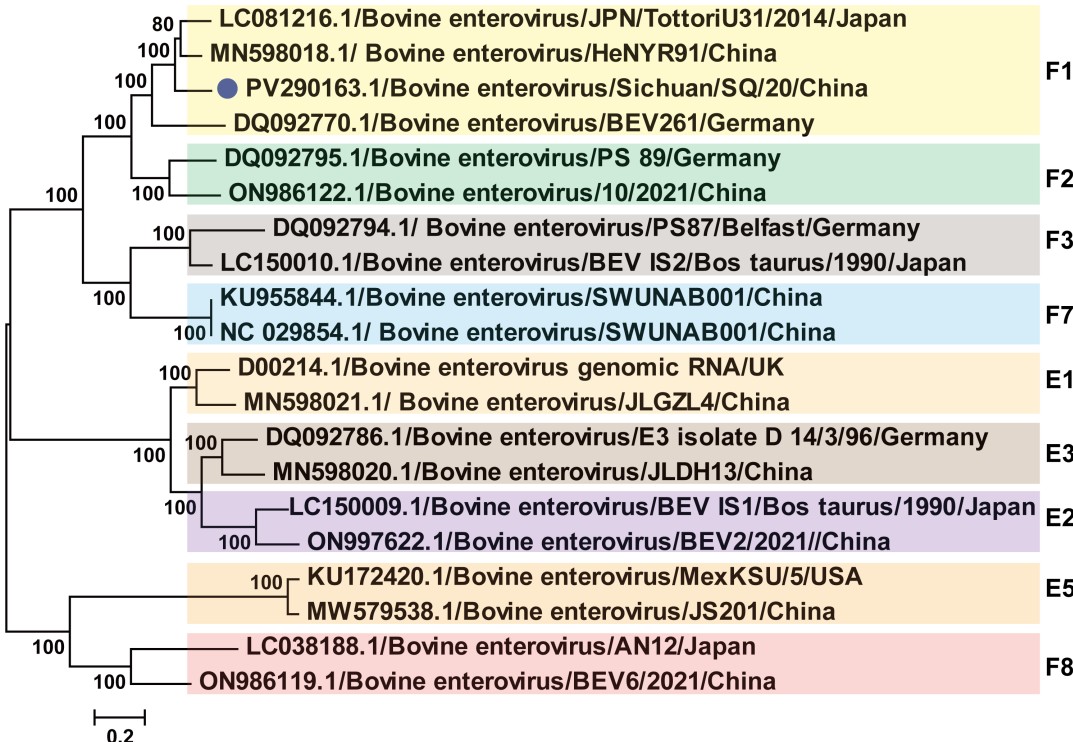

**FIG 3** Phylogeny of VP1 genomic nucleotide sequences. Bootstrap values were reconstructed by using the ML method available within the MEGA version 7.0 under GTR+Γ + I model with 1,000 replicates of the alignment, and only bootstrap values >70% are shown at appropriate nodes. The sequences determined in this study are marked with a blue circle.

## Analysis of genomic sequences

Full-length genomes of the isolated strains were amplified by RT-PCR with four pairs of primers (Table 1), and the PCR products of the expected size were obtained. The full-length genome sequence of the isolated strain (7,303 bp) was successfully assembled from fragmented sequencing data. It contains a single large ORF spanning 6,501 nucleotides, predicted to encode a 2,167-amino acid polyprotein.

## Phylogenetic analysis of BEV

To further investigate the phylogenetic relationships of Sichuan/SQ/20 strain, phylogenetic analysis was reconstructed based on the P1 and VP1 coding sequence. We performed a comparative analysis of VP1 nucleotide sequences from 20 representative strains. The phylogenetic analysis revealed that Sichuan/SQ-20, HeN-YR91, and JPN/TottoriU-31 clustered within the same branch with high bootstrap support, indicating their closest genetic relationship (Fig. 3). Phylogenetic analysis and amino acid sequence alignment of the P1 region from the Sichuan/SQ/20 strain further confirmed the aforementioned evolutionary relationships, showing consistent clustering patterns with other BEV-F1 subtype strains (Fig. 4).

To elucidate the genetic characteristics of the Sichuan SQ/20 strain, we performed a systematic phylogenetic analysis of the nucleotide and amino acid sequences of its P1 region in comparison with representative BEV reference strains. Nucleotide sequence similarity analysis revealed that Sichuan SQ/20 shares 61.8%–82.6% similarity with other strains in the P1 region, with the highest value (82.6%) observed against HeNYR91/China. At the amino acid level, the sequence identity between Sichuan SQ/20 and the reference strains ranged from 54.2% to 83.0% in the P1 region. Furthermore, Sichuan SQ/20 shares 55.6%–83.9% similarity with other strains in the VP1 region at the nucleotide level, with the highest value (83.9%) observed against HeNYR91/China (Table 2).

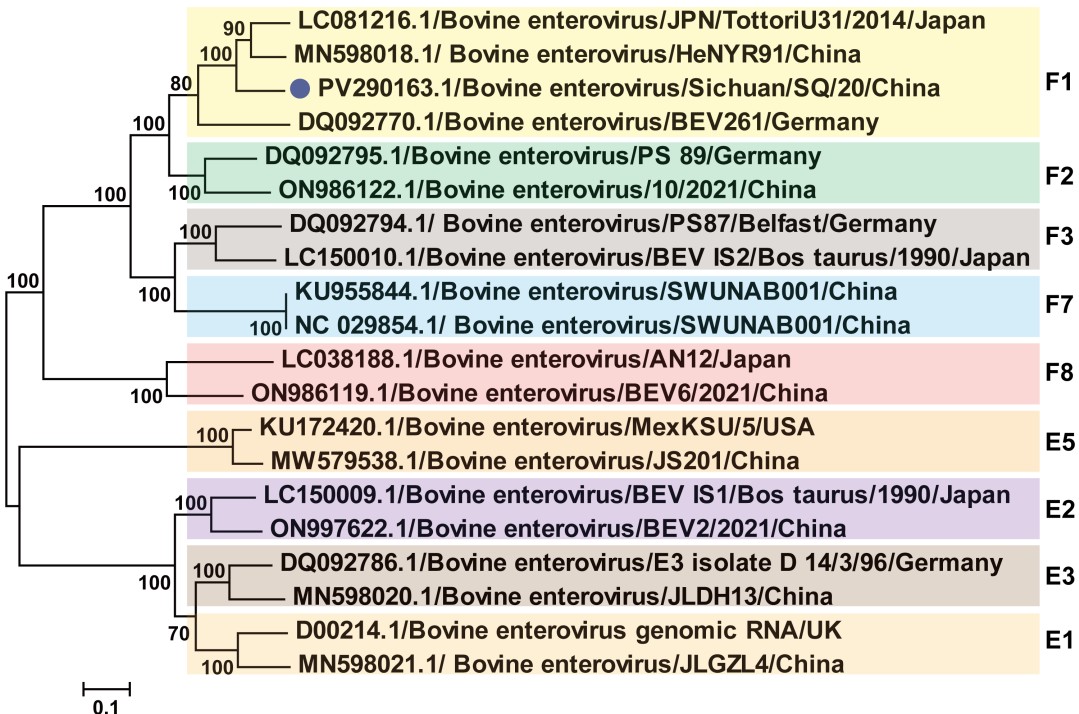

**FIG 4** Phylogeny of P1 structural polyprotein. Bootstrap values reconstructed as Fig. 3. The blue dot indicates the sequences determined in this study.

## Pathological lesions and viral tropism of Sichuan/SQ/20 in mice

Neither the experimental group nor the control group exhibited overt clinical signs during the study period. The tissue tropism of Sichuan/SQ/20 was determined by PCR analysis. Viral was detected in the spleen, liver, and small intestine at 3 days post-infection, with the highest viral load observed in the liver (as indicated by the brightest band on agarose gel electrophoresis; Fig. 5A). This pattern suggests that the liver may be a primary site of viral replication during early infection stages. However, at 7 days post-infection, the virus was detected exclusively in the spleen, while the liver and intestinal tissues tested negative (Fig. 5B), suggesting a clearance of the virus from peripheral sites and potential persistence within the splenic microenvironment. Tissue tropism was further confirmed by immunofluorescence detection. The results revealed specific fluorescent signals across all examined organ sections, indicating successful viral infection in these tissues (Fig. 5D), which was consistent with PCR results (Fig. 5A).

Necropsy findings revealed splenomegaly with dark red discoloration in mice, while no remarkable gross lesions were observed in other tissues. Histopathological examination of hematoxylin-eosin (H&E)-stained sections demonstrated red pulp atrophy, sinusoidal dilation, white pulp expansion, and lymphoid follicular hyperplasia in the spleen (Fig. 5C). In contrast, the liver and small intestine have no significant pathological alterations. These findings indicate that the spleen is the primary target organ of BEV infection.

Integrated analysis of histopathological lesions and viral distribution patterns indicates that this strain possesses limited pathogenicity, does not establish long-term latency or cause critical organ damage, undergoes a relatively short replication cycle, and can be effectively contained by the host immune system prior to widespread dissemination. These characteristics suggest that even though the virus can establish initial infection, it poses a relatively low risk for severe disease outcomes in immunocompetent hosts.

**TABLE 2** Analysis of nucleotide and amino acid sequence identity in the P1 region of Sichuan/SQ/20 and reference strains

| Reference strain | Accession no. | P1 | | VP1 | Type |
|---|---|---|---|---|---|
| | | nt (%) | aa (%) | nt (%) | |
| HeNYR91/China | MN598018.1 | 82.6 | 83.0 | 83.9 | BEV-F1 |
| JPN/TottoriU31/2014/Japan | LC081216.1 | 81.9 | 82.2 | 82.8 | BEV-F1 |
| BEV261/Germany | DQ092770.1 | 75.5 | 73.7 | 76.1 | BEV-F1 |
| 10/2021/China | ON986122.1 | 73.8 | 73.3 | 74.2 | BEV-F2 |
| PS 89/Germany | DQ092795.1 | 72.7 | 72.6 | 74.4 | BEV-F2 |
| IS2/Bos taurus/1990/Japan | LC150010.1 | 70.9 | 68.8 | 69.4 | BEV-F3 |
| PS87/Belfast/Germany | DQ092794.1 | 70.2 | 67.8 | 67.9 | BEV-F3 |
| SWUNAB001/China | NC_029854.1 | 69.7 | 66.8 | 68.0 | BEV-F7 |
| SWUNAB001/China | KU955844.1 | 69.7 | 66.8 | 68.0 | BEV-F7 |
| BEV6/2021/China | ON986119.1 | 63.1 | 58.8 | 59.3 | BEV-F8 |
| AN12/Japan | LC038188.1 | 63.8 | 58.7 | 58.3 | BEV-F8 |
| genomic RNA/UK | D00214.1 | 62.5 | 57.4 | 58.7 | BEV-E1 |
| JLGZL4/China | MN598021.1 | 61.9 | 56.8 | 57.9 | BEV-E1 |
| IS1/Bos taurus/1990/Japan | LC150009.1 | 63.3 | 58.9 | 59.6 | BEV-E2 |
| BEV2/2021/China | ON997622.1 | 62.6 | 56.9 | 57.8 | BEV-E2 |
| D 14/3/96/Germany | DQ092786.1 | 61.6 | 56.8 | 56.8 | BEV-E3 |
| JLDH13/China | MN598020.1 | 61.9 | 56.6 | 56.3 | BEV-E3 |
| MexKSU/5/USA | KU172420.1 | 62.4 | 55.1 | 56.0 | BEV-E5 |
| JS201/China | MW579538.1 | 61.8 | 54.2 | 55.6 | BEV-E5 |

## DISCUSSION

Cell line susceptibility is crucial for virus isolation. Studies have reported that BEV can be isolated using both MDBK and Vero cells (18, 21, 24). Specifically, Chengyuan Ji et al. demonstrated efficient BEV replication in MDBK cells, with lower propagation efficiency observed in BHK-21 and Vero cell lines (7). In this study, the virus isolation process began with screening 108 bovine fecal samples from four regions by PCR to identify positive cases. Virus isolation was simultaneously attempted from PCR-positive samples using MDBK, Vero, and BHK-21 cell lines. After five blind passages in each cell type, a novel bovine enterovirus strain capable of inducing stable CPE was successfully isolated in MDBK cells, designated as Sichuan/SQ/20. The isolated virus was subsequently subjected to whole-genome sequencing, and the complete nucleotide sequence was deposited into GenBank under accession number PV290163.1.

The VP1 region, serving as the primary antigenic determinant of the viral capsid, is internationally recognized as the gold standard for enterovirus typing, with >25% divergence indicating distinct types (25, 26). According to the International Committee on Taxonomy of Viruses standards, >60% amino acid sequence similarity in the P1 region is required for classification within the same enterovirus species (27). Phylogenetic analysis was performed following the classification criteria. Phylogenetic analysis of the amplified VP1 nucleotide and P1 amino acid sequences was conducted using the maximum likelihood method. The results demonstrated that this strain exhibits the closest genetic relationship with the HeN-YR91 and JPN/TottoriU-31 strains, and all three belong to the BEV-F1 genosubtype.

To elucidate pathogen characteristics, facilitate vaccine and diagnostic reagent development, and inform control strategies, the physicochemical and biological properties of the isolated strain were comprehensively analyzed. Analysis revealed that viral replication of the strain commenced at 3 hpi in MDBK cells, reaching peak titers at 48 hpi with a virus titer of $1 \times 10^{8.73}$ TCID$_{50}$/0.1 mL. CPE initially appeared at 12 hpi. IFA detection demonstrated virus-specific signals localized in the cytoplasm, confirming that the viral replication cycle was completed within the cytoplasmic compartment. Inactivation trials at different temperatures confirmed that the virus can maintain

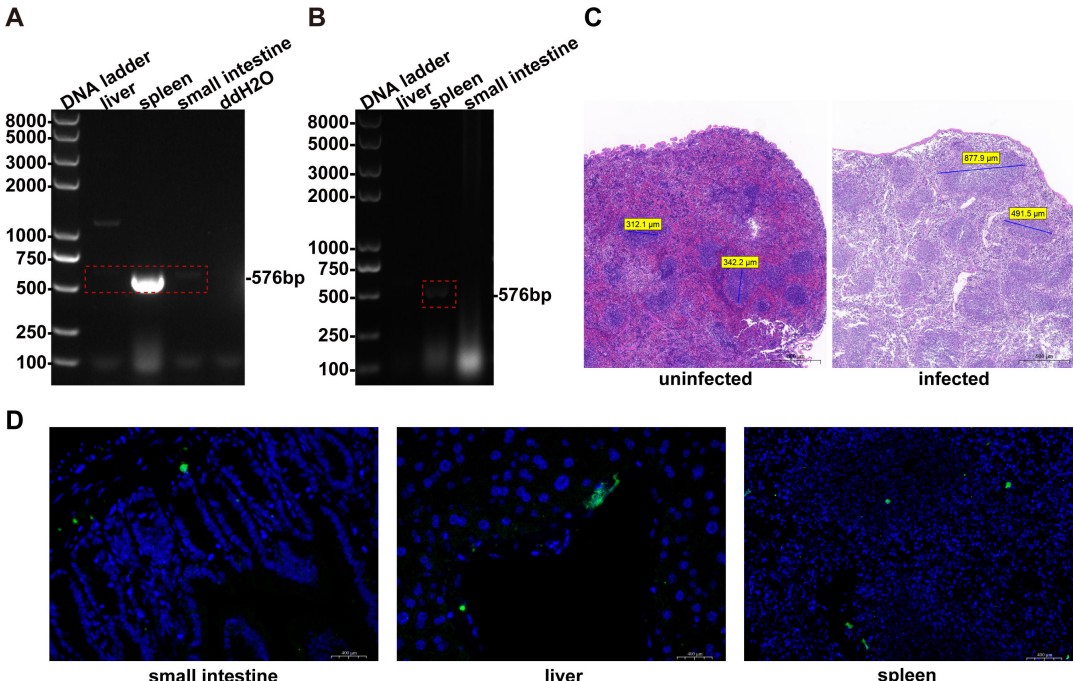

**FIG 5** Tissue tropism of Sichuan/SQ/20 in BALB/c mice. (A) PCR-based tropism profiling in visceral organs at 3 dpi (days post-infection). (B) PCR-based tropism profiling in visceral organs at 7 dpi. (C) Splenic histopathology following Sichuan/SQ/20 infection. Blue lines: anatomical width of the splenic white pulp. Scale bar, 500 µm. (D) IFA analyzed the distribution of viral nucleic acids in the small intestine, liver tissue, and spleen tissue sections of the mice challenged with Sichuan/SQ/20. The virus was stained with mAb 14D9 (green), and the nucleus was stained by DAPI (blue). Scale bar, 400 µm.

long-term viability at −80°C, analogous to entering a "dormant state." This finding has significant implications for establishing biosample handling protocols and investigating viral structure. However, since such extremely low temperatures are uncommon in daily environments, greater attention should be given to viral behavior under ambient and elevated temperatures. Complete inactivation of this strain after 12-minute exposure to 55°C, indicating thermal susceptibility consistent with other isolated strains (7).

To characterize the pathogenicity of the Sichuan/SQ/20 strain in mice, 6-week-old mice were infected with the strain, and samples were collected on days 3 and 7 post-infection for analysis. PCR, H&E staining, and IFA revealed viral detection in the liver, spleen, and small intestine at day 3 post-infection. At day 7, the virus was only detected in the spleen, indicating a gradual decline in viral load over time. H&E staining revealed significant pathological changes specifically in the spleen, with no lesions observed in other organs. IFA detected specific fluorescent signals in the liver, spleen, and small intestine, confirming viral infection in these tissues, which was consistent with PCR results. These findings indicate low susceptibility of mice to this virus, a self-limiting infection pattern, and distinct splenic tropism—collectively explaining the absence of notable clinical manifestations. Due to limitations of the current murine model system, further infection studies in other species are required to delineate the viral host range and infection mechanisms.

## Conclusion

In summary, this study successfully isolated a novel bovine enterovirus strain and conducted a comprehensive investigation into the physicochemical and biological properties of the Sichuan/SQ/20 isolate. These findings provide a scientific foundation

for vaccine design and antiviral drug screening, as well as the development of effective prevention and control strategies.

## ACKNOWLEDGMENTS

This work was supported by the Henan key research and development program (241111110300).

K.X.: investigation, validation, methodology, and writing—original draft. X.W.: validation. J.Y.G.: methodology. Y.K.: software. J.W.: formal analysis. B.C.: data curation. J.P.: writing review and editing. G.Y.: conceptualization, supervision, resources, and funding acquisition. All authors have read and agreed to the final manuscript.

## AUTHOR AFFILIATIONS

[1]College of Veterinary Medicine, Henan Agricultural University, Zhengzhou, Henan Province, China
[2]Key Laboratory of Animal Biochemistry and Nutrition, Ministry of Agriculture and Rural Affairs of the People's Republic of China, Zhengzhou, China
[3]Key Laboratory of Veterinary Biotechnology of Henan Province, Zhengzhou, China
[4]Henan University of Animal Husbandry and Economy, Zhengzhou, China

## AUTHOR ORCIDs

Kun Xu http://orcid.org/0009-0002-4624-2323
Beibei Chu http://orcid.org/0000-0003-2961-4754
Jiajia Pan http://orcid.org/0009-0006-3068-883X
Guoyu Yang http://orcid.org/0000-0003-2611-1588

## FUNDING

| Funder | Grant(s) | Author(s) |
| --- | --- | --- |
| Henan Key research and development program | 241111110300 | Guoyu Yang |

## AUTHOR CONTRIBUTIONS

Kun Xu, Investigation, Methodology, Validation, Writing – original draft | Xiaohan Wang, Validation | Jie Yuan Guo, Methodology | Yanpei Ku, Software | Jiang Wang, Formal analysis | Beibei Chu, Data curation | Jiajia Pan, Writing – review and editing | Guoyu Yang, Conceptualization, Funding acquisition, Resources, Supervision

## DATA AVAILABILITY

All data and materials generated for this study are included in the article. The nucleotide sequence of the isolated virus was deposited into GenBank under accession number PV290163.1.

## ETHICS APPROVAL

All the animals received humane care in compliance with good animal practice according to the animal ethics procedures and guidelines of the Institutional Animal Care and Use Committee (IACUC). All efforts were made to alleviate and minimize animal suffering.

## ADDITIONAL FILES

The following material is available online.

## Open Peer Review

**PEER REVIEW HISTORY (review-history.pdf).** An accounting of the reviewer comments and feedback.

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
