## [Reviewer comments · Microbiology Spectrum]

Microbiology Spectrum

Isolation and Identification of a Genotype F Bovine enterovirus in Western China

Kun Xu, Xiaohan Wang, Jieyuan Guo, Yanpei Ku, Jiang Wang, Bei-Bei Chu, Jiajia Pan, and Guo-Yu Yang

Corresponding Author(s): Guo-Yu Yang, College of Animal Sciences and Veterinary Medicine

Review Timeline:

Submission Date:	September 2, 2025
Editorial Decision:	October 14, 2025
Revision Received:	October 23, 2025
Accepted:	October 29, 2025

Editor: Peter Pelka

Reviewer(s): Disclosure of reviewer identity is with reference to reviewer comments included in decision letter(s). The following individuals involved in review of your submission have agreed to reveal their identity: Eduardo Rodríguez-Román (Reviewer #2)

Transaction Report:

DOI: <https://doi.org/10.1128/spectrum.02711-25>

Re: Spectrum02711-25 (Virome Analysis for Identification of a bovine enterovirus Isolated in Western China)

Dear Prof. Jia-Jia Pan:

Thank you for the privilege of reviewing your work. Below you will find my comments, instructions from the Spectrum editorial office, and the reviewer comments.

The reviewers found your manuscript meritorious but have requested some revisions that need to be made prior to publication. The MSA files for phylogenetic tree generation must also be made available.

Revision Guidelines

Sincerely,
Peter Pelka
Editor
Microbiology Spectrum

Reviewer #1 (Comments for the Author):

In this study, Xu et al describe the isolation and in vitro and in vivo characterization of a bovine enterovirus using a variety of complementary methods. Their studies are designed well, interpreted fairly and communicated clearly. While the impact of this identification remains to be seen, this report makes a worthwhile contribution to the scientific literature. I note the following minor feedback:

1. Title: the use of the word "virome" seems misplaced: this study describes the isolation and characterisation of a single virus
2. Fig 2a: define acronyms (eg hpi) within the Figure legend
3. Fig 5a/b: labeling of organ sites near the gel would make for easier reading
4. Fig 5c: legend should annotate what is being measured in the figure
5. Fig 5d: are all 3 columns necessary? Perhaps showing "merge" column alone would suffice?
6. Discussion: "Serological testing revealed serum antibody titers of approximately 1:400 in mice at both 3 and 7 days post-infection, indicating successful antibody production and activation of immune defense mechanisms following viral challenge." Are these data shown somewhere? They should be at described more fully in the Results section.

Reviewer #2 (Comments for the Author):

I would like to congratulate all the co-authors for their efforts in carrying out this work and completing this manuscript for review. I have several questions.

First, it is important to explain the differences between the bovine enteroviruses under study. According to what you have stated, the enteroviruses of interest are enterovirus E and enterovirus F, which are taxonomically classified as members of the species *Enterovirus eibovi* and *Enterovirus fitauri*, respectively.

According to the results you present, the SiChuan/SQ/20 isolate is classified within the F1 group of bovine enteroviruses, as are the HeN isolates YR91 and JPN/TottoriU. However, the latter are classified within the species *Enterovirus fitauri* and *Enterovirus eibovi*, respectively. The SiChuan/SQ/20 (PV290163.1) isolate is also classified within the species *Enterovirus eibovi*.

This difference should be clarified. The ICTV established that all enteroviruses previously designated enterovirus E (1-5) would be reclassified within the species *Enterovirus eibovi*, and those designated enterovirus F (1-8) would be reclassified within the species *Enterovirus fitauri*.

Regarding this, I would also like to know the reason for creating the phylogeny using a nucleotide MSA for the VP1 region? This phylogeny looks disorganized.

Why not create the phylogeny with the whole genome or by concatenated regions?

The phylogeny with the amino acid MSA for the P1 region looks clearly better, and this region already includes VP1.

Regarding the MEGA program, I would like to know why you used two different versions of the same program: version 7.0 and version X. Why use version 7.0 to create the tree, and then version X to create pseudoreplics? It doesn't make sense.

I think it's important to include a table showing the differences/similarities at the nucleotide and amino acid levels, by genomic regions, between the SiChuan/SQ/20 isolate and the other enteroviruses considered in this study, and compare these values with the ICTV taxa demarcation criteria for enteroviruses.

There are several details in the methodology. For example, it's important to mention that the primers used in this study were designed by you (if that's the case). In section 2.1, which refers to RT-PCR, it's not clear what methodology you used to perform cDNA synthesis. Perhaps the same as in section 2.8?

Once isolated, why didn't you inoculate the SiChuan/SQ/20 isolate into Vero and BHK-21 cells? This step is important to meet the virus's specificity. It's not clear whether you did this or not.

In Figure 1D, you refer to a scale bar of 100 nm, but this bar does not appear in the image.

You have a long and complete work, and I think the text could be improved to make the information clearer.

Good luck, guys.

Reviewer #1 (Comments for the Author):

In this study, Xu et al describe the isolation and in vitro and in vivo characterization of a bovine enterovirus using a variety of complementary methods. Their studies are designed well, interpreted fairly and communicated clearly. While the impact of this identification remains to be seen, this report makes a worthwhile contribution to the scientific literature. I note the following minor feedback:

1. Title: the use of the word "virome" seems misplaced: this study describes the isolation and characterisation of a single virus

Response: Thanks for your valuable suggestion. We have corrected the title into "Isolation and Identification of a Genotype F Bovine enterovirus in Western China" according to your suggestion.

2. Fig 2a: define acronyms (eg hpi) within the Figure legend

Response: Thanks for your correction. We felt sorry that we didnot provide enough information aout the manuscript in detail. We have defined acronyms used in Figure 2A (hpi: hours post infection) and Figure 5A (dpi: days post infection) within the figure legends according to your suggestion.

3. Fig 5a/b: labeling of organ sites near the gel would make for easier reading

Response: Thanks for your suggestion. We have labeled the organ sites near the gel in Figures 5A and 5B, and have also optimized the corresponding figure legends.

4. Fig 5c: legend should annotate what is being measured in the figure

Response: Thanks for your valuable suggestion. In accordance with your suggestion, we have annotated the measured parameters in Figure 5C.

5. Fig 5d: are all 3 columns necessary? Perhaps showing "merge" column alone would suffice?

Response: Thanks for your valuable comment. According to your suggestion, we removed the "DAPI" and "mAb 14D9" columns, showing only the "merge" column in Figure 5D. Additionally, we added the scale bar in Figure 5D.

6. Discussion: "Serological testing revealed serum antibody titers of approximately 1:400 in mice at both 3 and 7 days post-infection, indicating successful antibody production and activation of immune defense mechanisms following viral challenge." Are these data shown somewhere? They should be at described more fully in the Results section.

Response: Thanks for your valuable comments sincerely. We fully acknowledge that serological antibody titers alone are insufficient to conclusively demonstrate the successful activation of the host's immune defense mechanisms following viral challenge. Accordingly, we have removed all related serological descriptions from the manuscript as suggested. This important insight has provided clear direction for our future research, wherein we plan to undertake more systematic experiments, such as assays for cellular immune responses, for further validation. Once again, we extend our sincere gratitude for your rigorous and valuable feedback.

Reviewer #2 (Comments for the Author):

I would like to congratulate all the co-authors for their efforts in carrying out this work and completing this manuscript for review.

I have several questions.

1. First, it is important to explain the differences between the bovine enteroviruses under study. According to what you have stated, the enteroviruses of interest are enterovirus E and enterovirus F, which are taxonomically classified as members of the species *Enterovirus eibovi* and *Enterovirus fitauri*, respectively. According to the results you present, the SiChuan/SQ/20 isolate is classified within the F1 group of bovine enteroviruses, as are the HeN isolates YR91 and JPN/TottoriU. However, the latter are classified within the species *Enterovirus fitauri* and *Enterovirus eibovi*, respectively. The SiChuan/SQ/20 (PV290163.1) isolate is also classified within the species *Enterovirus eibovi*. This difference should be clarified.

Response: Thanks for your suggestions. The phylogenetic tree was constructed from actual sequence alignments, following genotyping against the BEV261/Germany reference strain (DQ092770.1). The SiChuan/SQ/20 isolate is classified within the F1 group of bovine enteroviruses, as are the HeN isolates YR91 and JPN/TottoriU. Actually, the genotype of the JPN/TottoriU strain is classified as F group of bovine enteroviruses according to the International Committee on Taxonomy of Viruses (ICTV) standards. However, we found that the genotype of the JPN/TottoriU strain differed from the description record in the GenBank database, suggesting that this discrepancy may be due to a mismatch between its actual genetic background and the original description in GenBank.

2. The ICTV established that all enteroviruses previously designated enterovirus E (1-5) would be reclassified within the species *Enterovirus eibovi*, and those designated enterovirus F (1-8) would be reclassified within the species *Enterovirus fitauri*. Regarding this, I would also like to know the reason for creating the phylogeny using a nucleotide MSA for the VP1 region? This phylogeny looks disorganized. Why not create the phylogeny with the whole genome or by concatenated regions? The phylogeny with the amino acid MSA for the P1 region looks clearly better, and this region already includes VP1.

Response: Thanks for your suggestion. Studies have confirmed that the VP1 sequence serves as an effective tool for EV genotyping. Accordingly, we firstly performed phylogenetic analysis based on this major antigenic determinant for preliminary genotyping identification. Subsequently, in accordance with the ICTV species demarcation criterion (P1 region amino acid sequence divergence <40%), a confirmatory analysis was conducted using the complete P1 region sequence (which includes VP1) to achieve precise species classification.

3. Regarding the MEGA program, I would like to know why you used two different versions of the same program: version 7.0 and version X. Why use version 7.0 to create the tree, and then version X to create pseudoreplicas? It doesn't make sense.

Response: Thanks for your suggestion. In fact, we only used MEGA 7.0 to build the phylogenetic tree and pseudoreplicas. The "version X" was wrong typed in the submitted manuscript. We have corrected the version description in the Methods section (Section 2.9).

4. I think it's important to include a table showing the differences/similarities at the nucleotide

and amino acid levels, by genomic regions, between the SiChuan/SQ/20 isolate and the other enteroviruses considered in this study, and compare these values with the ICTV taxa demarcation criteria for enteroviruses.

Response: Thanks for your valuable suggestion sincerely. We have incorporated an analysis of sequence identity (including both nucleotide and amino acid sequences) in the P1 region between SiChuan/SQ/20 and reference strains into Section 3.4 of the manuscript, with the corresponding results presented in Table 2.

Table 2 Analysis of nucleotide and amino acid sequence identity in the P1 region of SiChuan/SQ/20 and reference strains

		SiChuan/SQ/20 Strains identity			
Reference strains	Accession no.	P1		VP1	type
		nt (%)	aa (%)	nt (%)	
HeNYR91/China	MN598018.1	82.6	83.0	83.9	BEV-F1
JPN/TottoriU31/2014/Japan	LC081216.1	81.9	82.2	82.8	BEV-F1
BEV261/Germany	DQ092770.1	75.5	73.7	76.1	BEV-F1
10/2021/China	ON986122.1	73.8	73.3	74.2	BEV-F2
PS 89/Germany	DQ092795.1	72.7	72.6	74.4	BEV-F2
IS2/Bos taurus/1990/Japan	LC150010.1	70.9	68.8	69.4	BEV-F3
PS87/Belfast/Germany	DQ092794.1	70.2	67.8	67.9	BEV-F3
SWUNAB001/China	NC_029854.1	69.7	66.8	68.0	BEV-F7
SWUNAB001/China	KU955844.1	69.7	66.8	68.0	BEV-F7
BEV6/2021/China	ON986119.1	63.1	58.8	59.3	BEV-F8
AN12/Japan	LC038188.1	63.8	58.7	58.3	BEV-F8
genomic RNA/UK	D00214.1	62.5	57.4	58.7	BEV-E1
JLGZL4/China	MN598021.1	61.9	56.8	57.9	BEV-E1
IS1/Bos taurus/1990/Japan	LC150009.1	63.3	58.9	59.6	BEV-E2
BEV2/2021/China	ON997622.1	62.6	56.9	57.8	BEV-E2
D 14/3/96/Germany	DQ092786.1	61.6	56.8	56.8	BEV-E3
JLDH13/China	MN598020.1	61.9	56.6	56.3	BEV-E3
MexKSU/5/USA	KU172420.1	62.4	55.1	56.0	BEV-E5
JS201/China	MW579538.1	61.8	54.2	55.6	BEV-E5

5. There are several details in the methodology. For example, it's important to mention that the primers used in this study were designed by you (if that's the case). In section 2.1, which refers to

RT-PCR, it's not clear what methodology you used to perform cDNA synthesis. Perhaps the same as in section 2.8?

Response: Thanks for your suggestions. We felt sorry that we did not provide enough information in the methodology. We added the details about the experiments in the method that were marked in red. The primers used in this study were shown in Table 1. Additionally, this study employed the Vazyme ClonExpress II One Step Cloning Kit (C112-01) (Vazyme Biotech Co., Ltd.) for target gene amplification, achieving highly efficient amplification.

Once isolated, why didn't you inoculate the SiChuan/SQ/20 isolate into Vero and BHK-21 cells? This step is important to meet the virus's specificity. It's not clear whether you did this or not.

Response: Thanks for your suggestions. In the virus isolation experiment described in Section 2.3, the same BEV-positive sample filtrate induced cytopathic effects (CPE) exclusively in MDBK cells, while no changes were observed in Vero or BHK-21 cells. Given that all three cell lines were inoculated in parallel and only MDBK cells exhibited specific CPE, these results indicate that the virus possesses a specific tropism for MDBK cells.

6. In Figure 1D, you refer to a scale bar of 100 nm, but this bar does not appear in the image. You have a long and complete work, and I think the text could be improved to make the information clearer.

Response: We felt sorry for our carelessness. We have added the scale bar into Figure 1D, which allowed accurate size interpretation of the results.

Re: Spectrum02711-25R1 (**Isolation and Identification of a Genotype F Bovine enterovirus in Western China**)

Dear Dr. Kun Xu:

Your manuscript has been accepted, and I am forwarding it to the ASM production staff for publication. Your paper will first be checked to make sure all elements meet the technical requirements. ASM staff will contact you if anything needs to be revised before copyediting and production can begin. Otherwise, you will be notified when your proofs are ready to be viewed.

Sincerely,
Peter Pelka
Editor
Microbiology Spectrum